# Changing Trends in Paralytic Shellfish Poisonings Reflect Increasing Sea Surface Temperatures and Practices of Indigenous and Recreational Harvesters in British Columbia, Canada

**DOI:** 10.3390/md19100568

**Published:** 2021-10-14

**Authors:** Lorraine McIntyre, Aroha Miller, Tom Kosatsky

**Affiliations:** Environmental Health Services, British Columbia Centre for Disease Control, 655 West 12th Avenue, Vancouver, BC V5Z 4R4, Canada; arohamiller@gmail.com (A.M.); tom.kosatsky@bccdc.ca (T.K.)

**Keywords:** climate change, harmful algal blooms, indigenous, marine toxins, paralytic shellfish poisoning, saxitoxin, bivalves, public health

## Abstract

Paralytic shellfish poisoning (PSP) occurs when shellfish contaminated with saxitoxin or equivalent paralytic shellfish toxins (PSTs) are ingested. In British Columbia, Canada, documented poisonings are increasing in frequency based on 62 investigations identified from 1941–2020. Two PSP investigations were reported between 1941 and 1960 compared to 31 since 2001 (*p* < 0.0001) coincident with rising global temperatures (*r*^2^ = 0.76, *p* < 0.006). The majority of PSP investigations (71%) and cases (69%) were linked to self-harvested shellfish. Far more investigations involved harvests by indigenous communities (24%) than by commercial and recreational groups. Single-case-exposure investigations increased by more than 3.5 times in the decade 2011–2020 compared to previous periods. Clams (47%); mussels (26%); oysters (14%); scallops (6%); and, in more recent years, crabs (4%) were linked to illnesses. To guide understanding of self-harvesting consumption risks, we recommend collecting data to determine when PST-producing algae are present in high concentrations, improving the quality of data in online shellfish harvest maps to include dates of last testing; biotoxin testing results; and a description of bivalve species tested. Over reliance on toxin results in biomonitored species may not address actual consumption risks for unmonitored species harvested from the same area. We further recommend introducing phytoplankton monitoring in remote indigenous communities where self-harvesting is common and toxin testing is unavailable, as well as continuing participatory education about biotoxin risks in seafoods.

## 1. Introduction

Paralytic shellfish poisoning (PSP) is a global phenomenon, with illnesses reported in many countries [1,2,3,4,5,6,7,8]. In British Columbia (BC), Canada, PSP was first recorded in 1793 when bouts of severe illness and one death occurred in four members of Captain George Vancouver’s crew, following a meal of mussels harvested from Poison Cove in Nootka Sound on Canada’s Pacific coast [9].

PSP occurs when a mixture of saxitoxins, saxitoxin analogues (STX-eq), and related paralytic shellfish toxins (PSTs) are ingested in contaminated shellfish [10]. These phycotoxins are produced by marine eukaryotic dinoflagellate algae (*Alexandrium*, *Gymnodinium*, and *Pyrodiunium*), calcareous red macroalgae, and freshwater prokaryotic cyanobacteria (*Lyngbya wollei*, *Anabaena circinalis*, *Cylindrospermopsis raciborskii*, *Planktothrix* sp., *Aphanizomenon flos-aquae*, and others) [9,11,12]. When environmental conditions are optimal, algae can proliferate in events termed “harmful algal blooms” (HABs) [13,14]. The occurrence and geographical range of HABs is expanding [15], and longer seasonal windows for HABs in temperate-zone waters are expected [14,16]. Moreover, favorable conditions for toxin-producing algal species are expected to increase [5,13,17], mediated by climate change [16,18], linked to El Niño events [2], eutrophication [19], and anthropogenic activity [20]. Consequently, PSP illnesses are expected to rise, especially among people who themselves gather shellfish for consumption as opposed to among those who consume commercially harvested, inspected, and tested shellfish.

There are 58 or more PST neurotoxin alkaloid compounds. In humans, PSTs have also been shown to block muscle action potential and nerve conduction [11] by preventing passage of sodium ions through sodium gated channels. In muscle and neuron cells, this inhibits the generation of action potential, preventing muscle contractions [21]. Human illness occurs primarily through ingestion of species in which toxins have bioaccumulated, particularly in the digestive tracts of bivalve mollusks [22] but also in crabs, predatory gastropod species, and the adductor muscle of rock scallops [23]. PSP presents with both gastrointestinal and neurological manifestations, typically within 5 to 30 min of exposure. Initial symptoms include a tingling sensation around the mouth, followed by numbness in the fingers and/or toes [14]. Numbness can progress to all four limbs, the torso, and the neck. Respiratory muscle and diaphragm paralysis can occur in severe cases. Other symptoms include nausea, vomiting, diarrhea, dizziness, headache, loss of co-ordination, and dysphagia [24]. While mortality rates, due to respiratory paralysis, can reach as high as 15% when no intervention is available, patients recover if they survive the first 24 h [10,25]. Beyond supportive therapy, treatment involves facilitating toxin dilution, adsorption of toxins via administration of charcoal, and/or renal excretion of the toxin, with ventilation in the case of respiratory muscle paralysis [22]. There is no known antidote for PSTs [24]. Recovery occurs within hours in mild intoxications and over several days to weeks with large toxin dose ingestion.

PSTs cannot be eliminated from food by freezing or heating. Further, cooking can transform PST from a less toxic into a more toxic conformation [10]. Toxins present in digestive tissues (viscera or hepatopancreas) of seafoods may be released during boiling, becoming more concentrated in the broth and contaminating other foods cooked in the broth, including muscle tissues that would otherwise have low toxicity, such as scallop adductors, crab legs, and claws.

In Canada, a national monitoring program for bivalve mollusks for PSTs has been in place since 1942 [23]. Indicator samples, typically biomonitored mussels, and commercial species collected from harvesting areas are tested for the presence of shellfish biotoxins. Commercially sold shellfish are required to come from open, approved harvest areas that meet shoreline assessments, water quality criteria, and regulatory levels for shellfish biotoxins as specified in the Canadian Shellfish Sanitation Program (CSSP) [26]. A maximum marketable limit of 0.8 mg STX-eq/kg in bivalve shellfish edible tissue is maintained by Health Canada [27,28]. PSP was added to BC’s list of reportable diseases in 2001 (Schedule A—reportable by all sources) and has been on the list of Canada’s nationally notifiable diseases since 2007 [29].

Here, we present a longitudinal record of PSP illnesses and harvester demographics. We compare illnesses to harvest area status, examine recent trends of PSP investigations in BC to better understand which populations are most affected, and recommend actions on how to reduce potential exposures to protect public health.

## 2. Results

### 2.1. Investigation Trends

We collated reports of 62 BC PSP investigations between 1940 and 2020. Over one-third (35%) of all investigations took place within the period 2011 to 2020, Figure 1a, with numbers of investigations increasing significantly each 20-year period (*χ*^2^ = 31.7, *p* < 0.001). There were 36 (58%) PSP investigations of ≥2 cases and 26 (42%) individual exposure events. Larger PSP outbreaks were reported more frequently in earlier decades with smaller clusters and single-case exposures reported through-out the 80-year period. Most (54%) single-case exposures occurred in the period 2011 to 2020. Overall, 301 PSP cases and five deaths were recorded from 1940 to 2020 for a case fatality rate of 1.6%. The most recent death reported was in 1980. An additional 16 individuals associated with eight clusters were not ill after ingesting the implicated shellfish.

### 2.2. Case Demographics and Symptoms

PSP was reported more often in males (57%) than females (43%, Table 1, based on records by sex for 111 cases from 51 investigations). The median age of cases was 46 (mean 44) years, ranging from 8 to 91 years (based on 61 cases from 38 investigations). The median symptom onset was 90 min (mean 3.2 h), ranging from 5 min to 18 h (based on 97 cases from 51 investigations). Duration of PSP illness from onset varied from 30 min to 10.5 days with a median duration of 24 h and mean duration of 38 h until hospital discharge (based on 102 cases from 52 investigations). Three cases were hospitalized for four or more days. Deaths occurred between 3.5 and 8 h after onset in four cases. Nearly half of the cases (49%) were seen in the emergency room or hospital; others were seen by a physician outside of hospital (8%) or required ambulance or medi-vac assistance (3%).

The majority of PSP illnesses (69%) and investigations (71%) were linked to self-harvested shellfish, predominantly gathered by indigenous harvesters (22% cases, 24% investigations), followed by local community harvesters (49% cases, 14% investigations); recreational harvesters identified as boaters, campers, or tourists (15% cases, 21% investigations); and commercial fishers (4% cases, 10% investigations), shown in Figure 1b and Figure 2. A single investigation in 1957 of 111 cases attributed to local harvesters represented 39% of all cases within this group. All recorded deaths occurred in indigenous harvesters. While illness in indigenous harvesters and commercial fishers was present throughout the 80-year record, increasing numbers of recreational self-harvesters, such as campers, boaters, and local non-indigenous harvesters, were reported in the last decade. Identity of self-harvesters could not be determined in 11 investigations. PSP investigations linked to commercially sourced shellfish were from restaurants (14%) or retail stores (19%), including one farmers’ market. Prior to 1980, no restaurant or retail sources of the implicated shellfish were identified; equal numbers were identified in the last decade (2011–2020, *n* = 9) when compared to the previous three decades (1981–2010, *n* = 9).

The most common symptoms (recorded in 189 cases from 57 investigations, Table 1) were tingling in the mouth, face, and tongue (74%); and hands and feet (46%), followed by numbness (68%), perioral numbness (21%), numbness in hands and feet (20%), incoordination or ataxic gait (43%), partial to complete paralysis (18%), abdominal cramping and discomfort (15%), sensations of floating or dizziness (14%), nausea and/or vomiting (14%), weakness (10%), and diarrhea (7%). Other symptoms included headache (6%); difficulty breathing (4%); swelling of lips, tongue, or facial area (3%); chest pain or rapid pulse (3%); dysphagia or dysarthria (3%); ptosis (2%); intubation (2%); and loss of consciousness (2%). In three investigations, cases who reported vomiting experienced decreased symptom severity in comparison to other cases within the same cluster, such as numbness without progression to ataxia and shorter symptom duration.

The recording of case details changed with the introduction of standardized forms (2003) and the implementation of data-bases (2011), resulting in both improvements and deficits, Figure 1c [30,31]. Completeness of symptom information deteriorated when a generic approach to classifying shellfish illness as either “clinical gastroenteritis (vomiting, diarrhea)” or “neurological symptoms (numbness, tingling sensation)” occurred in standardized forms using a check-box format, rather than recording individual specific symptoms for each case. Data-base prompts to enter source and location of shellfish, such as harvest areas implicated, improved the overall level of detail included with each case. Over the 80-year period, prior to 1990, most investigation information was found in published articles and newsletters. Major report sources transitioned over the next 20 years to interagency faxes, e-mails, letters, newspaper articles, poison control paper records, and DFO online harvest area information. Since 2011, investigation reports from poison control surveillance subsequently recorded in databases formed the majority of BC PSP reports (74%).

### 2.3. Food Sources and Toxin Dose

Illness was most often associated with clams (47%), followed by mussels (26%), oysters (14%), scallops (6%), crabs (4%), and cockles (3%). Consumption of more than one type of seafood was noted in 13% of investigations. Among clam species, butter clams were most often identified (42%), followed by manila and little neck clams (6%), with no species recorded in 45% of investigations (Figure 1d). Self-harvested crab was identified eaten with commercially sourced oysters (in 1991) and twice as the only implicated food (in 2006 and 2017).

Toxin values exceeded 1000 µg STX-eq/100 g in over one-third of investigations, appearing more frequently prior to 1990, and exceeded 10,000 µg STX-eq/100 g on five occasions, with the highest value ingested being 20,000 µg STX-eq/100 g in 2019 (Figure 1e). Illnesses have been recorded when no toxin was detected in shellfish since 1980, although illness with no toxin detected has been noted more frequently (64%) in the last decade (2011–2020). Toxin data were available in 75% of investigations, from tests conducted in leftover shellfish (25%), shellfish collected from the same batch at retail store or restaurant premises (8%), or biomonitored shellfish (43%) collected from the same or adjacent harvest area; 10% of biomonitored results were not recent enough (±3 weeks) to implicate harvest area risk (Figure 1f). Mussels were the predominant biomonitored species (72%) but did not represent the shellfish consumed in 13% of PSP investigations. Toxin results were unavailable when the harvest area where shellfish were collected from was unknown (11%) or when there were no leftover shellfish available for testing and biomonitored samples were unavailable (11%). PST levels in shellfish consumed by cases are shown in Figure 3 (based on data for toxin values in leftovers and biomonitored shellfish from BC sources as described in Appendix A. Greater variability of toxin amounts was observed in mussel samples linked to illness, although median values of toxins observed were higher in clams (*n* = 17, 2200 µg STX-eq/100 g) over mussels (*n* = 10, 810 mussels µg STX-eq/100 g) and lowest in other shellfish species (*n* = 3, oysters and cockles).

### 2.4. Harvest Site Locations and Status of PSP Investigations

Harvest site location was mapped by demographic identity (who harvested, Figure 4a) and geo-temporally (when) for PSP illnesses over the 80-year period (Figure 4b). Shellfish implicated in illnesses were traced to 42 separate harvesting areas in BC, and 12 areas were identified in ≥2 investigations. Inlets located north of Powell River (Theodosia and Okeover) were implicated in four separate PSP investigations in harvest management area 15–4 on the north-eastern end of the Salish Sea [32]. Other locations included sites on the west coast and inner protected east coast of Vancouver Island, on Haida Gwaii Island, along exposed coastal and protected inlet areas. North of 50° latitude, all illnesses arose from self-harvested shellfish among indigenous, fishermen, and recreational groups. In lower latitudes, PSP illnesses linked to commercial sources were more concentrated in the northern region of the Georgia Straight, Salish Sea area. Self-harvested shellfish sources occurred along much of BC’s coast and throughout the Salish Sea in the Strait of Juan de Fuca, Georgia Straight, on Haida Gwaii (formerly known as Queen Charlotte Islands) and western Vancouver Island. Since 2011, there were 15 PSP investigations clustered within harvesting areas in the Salish Sea, and three occurred in each of northern BC, non-BC commercial sources, and from unknown sources. Overall, commercial shellfish sources were identified from BC (23%), Eastern Canada (5%, Nova Scotia, Newfoundland, and Prince Edward Island), and imports (3%, Washington State and Thailand) in PSP investigations. Multiple shellfish harvest locations were identified 11% of investigations, 7% of confirmed (i.e., PSTs detected in shellfish), and 3% of probable PSP investigations. Source location was unknown or unspecified for 12% of PSP investigations, occurring when self-harvest location was not disclosed by the case or when shellfish were purchased at a shop or consumed at a restaurant without trace back to a geographical region available in invoice or shellfish tag records. A summary of all investigations is shown in Appendix A.

We reviewed whether the shellfish harvesting area was open or closed at time of harvest (harvest area status) for each PSP investigation (Table 2). The status was known for 39 (89%) self-harvest associated and 10 (59%) commercially sourced shellfish investigations. Harvest areas were in an open status in 21% of self-harvest PSP investigations (11% confirmed and 10% probable) and in a closed status in 42% of self-harvest PSP investigations (21% of confirmed and 21% of probable). In contrast, no commercial harvest areas were in open status in confirmed PSP investigations linked to purchased shellfish. Two harvest areas were closed (3%); one of the investigations was linked to illegally harvested shellfish in 2003 (report #34, Appendix A) and the other transitioned from an open to closed status during a rapid increase in PST in the harvest area in 2011 (report #44, Appendix A). Remaining commercial shellfish linked to probable PSP illness investigations (13%) were in open harvest areas.

### 2.5. Seasonal Trends and Temperature

PSP occurred in all months of the year except February. May was the most common month for reported illnesses (*n* = 14). Differences were observed in the numbers of investigations by month (*χ*^2^ = 19.0, *p* = 0.001), with fewer observed in winter (January to March) over other seasons (*χ*^2^ = 17.5, *p* = 0.014). No difference was observed in the number of investigations that occurred in months with a ‘R’ (September to April) compared to months without an ‘R’ (May to August). Elevated PSTs in shellfish monitoring samples were more frequently present in warmer months (May–October) compared to the rest of the year (data not shown), although PSTs in shellfish tissue can exceed 0.8 mg STX-eq/kg in any month of the year in BC [33]. Increasing numbers of PSP shellfish investigations were coincident with increases observed in BC coastal sea surface temperatures (SST) (Figure 5, *r*^2^ = 0.74, *p* = 0.006), with SST increasing 0.7 °C from 1941 to present.

## 3. Discussion

This catalogue of PSP in BC, Canada included 36 investigation clusters involving 279 cases and 26 single-case exposure investigations over an 80 year span. Nearly half (48%) of all investigations occurred since 2002, associated with 54 PSP illnesses. A longitudinal PSP data set from Alaska spanning 20 years from 1973 to 1992 identified 54 outbreaks involving 117 illnesses [3], and, in 2010–2011, also reported single-case exposures [34]. Similar to Alaska, BC PSP cases were often hospitalized, admitted to emergency rooms, or visited by physicians outside of the hospital [3]. In 2011–2020, single-case exposures increased over 3.8 times when compared to the average number of single cases occurring in the previous 30 years (since 1991) and over 4.6 X when compared to cases since 1940. Regardless, outbreaks with one or more case exposures involve multiple agencies, are labor-intensive, and place a significant burden on the health care system.

Changing patterns of marine phycotoxins have been attributed to changing climate conditions. Increased ocean temperatures, loss of oxygen, acidification, and loss of density in upper ocean layers affecting stratification in deeper ocean layers are all observed in the North Pacific Ocean [18]. In our study, increasing numbers of PSP investigations are occurring coincident with rising global ocean temperatures. Similar to Alaska, Washington State, and the United Kingdom, PSP illnesses in BC occurred more frequently during warmer months (May through October) [3,35,36,37,38]. In BC, PSP illnesses occur most frequently in the month of May, prior to peak spring freshet—a mix of snow melt and fresh rainwater that typically occurs in the month of June. This observation is consistent with modelling that found PST events in biomonitored samples collected on the west coast of Vancouver Island, BC occur when freshwater discharge levels are lower than 200 m^3^ s^−1^ [33]. This model further predicts that PST frequency on the west coast is also driven by salinity and upwelling, in comparison to major shellfish harvesting areas inland that are additionally affected by air and sea temperature [33]. Over the last decade, more PSP investigations have occurred in the inland regions of the Salish Sea than in other BC coastal areas. Stable, warm SSTs favor dinoflagellate growth and harmful algal bloom formation [5,33,39]. Three dinoflagellate species, *Alexandrium acatanella*, *A. catenella*, and *A. ostenfeldii*, were identified in causing PST blooms in BC in recent decades, based on data collections in the harmful algae event database for the Pacific region. El Niño and climate-change-driven SST rises increase the occurrence of PSP in warmer months [2,7]. Our review did not find any difference in the frequency of PSP events in months with or without an ‘R’, negating the myth that shellfish may be safely harvested in months containing an ‘R’, although, seasonally, fewer PSP illnesses occurred in the winter months.

Indigenous harvests of shellfish account for nearly one quarter of all PSP outbreaks, even though indigenous peoples only account for 5.9% of the BC population [40]. PSP was documented in northern indigenous communities and in communities around Vancouver Island. Traditional self-harvesting activities often occur in remote communities underserviced by commercial testing for PSTs and where sample transport is limited. BC has a long coastline; many islands; inlets; and small, remote settlements. Self-harvest areas not covered by routine shellfish biomonitoring, because they fall outside of commercial harvest areas, put self-harvesters, particularly indigenous communities reliant on shellfish, at greater risk. Introduction of phytoplankton monitoring conducted by communities in these areas could serve as an early warning system for HABs, and access to rapid PST test kits would allow community shellfish harvests to be tested immediately prior to consumption improving safety. In this study, a review of harvest area status of indigenous self-harvests found that 43% occurred in open harvest areas and 46% occurred in closed harvest areas. Neighbouring US states (Washington, Alaska) have created phytoplankton monitoring programs within tribal communities to provide community knowledge of marine biotoxin risks [41], a step that needs to occur in BC. Illnesses in indigenous groups are likely under-estimated and may not have been reported for a variety of reasons, including a lack of (historical) reporting from indigenous communities, not wishing to self-identify as conducting community harvests, a lack of ready access to medical attention, and mistrust of government.

The majority (70%) of restaurant and retail store purchases of shellfish linked to PSP have occurred since 2003. Confirmed cases of PSP linked to consumers purchasing shellfish at retail or in restaurants are rare. Six confirmed cases of PSP in one BC outbreak were linked to commercially purchased shellfish from a retail store. The mussels consumed were harvested illegally, uninspected, and sold to the store through unregulated channels. This store and owner were criminally prosecuted and fined [42,43]. The only other confirmed commercial source of shellfish that caused PSP occurred with a single-case exposure in a restaurant. Shellfish were sourced from a harvest area where PSTs spiked in biomonitored mussel samples over a one-week period during a commercial harvest in October 2011. The mussels were harvested, while the harvest area was still in an open status before biomonitoring samples results were known. Mussels filter high volumes of water and thus accumulate toxins faster than other shellfish species [44]. These shellfish were subsequently recalled from the marketplace, likely preventing other illnesses [45]. Rapidly developing HABs occur with little warning, limiting the value of routine sampling at discrete time points, potentially resulting in more frequent shellfish product recalls. Rapid rises in shellfish biotoxin levels are a concern for the industry, regulators, and public health, with substantial costs for lost opportunities for commercial, community, and recreational fisheries [41].

The CSSP program has been largely successful in preventing PSP in most commercial shellfish sources. Since 2006, we identified 30 shellfish biotoxin-associated food recalls, with BC products accounting for 80% of these recalls. Recalls likely play a key role in preventing PSP illnesses from commercial sources; however, as evidenced by the PSP investigation reported here, biotoxin levels can increase rapidly. The need to issue recalls to prevent contaminated shellfish from reaching the consumer indicates that routine PST biomonitoring by itself, which typically occurs every one to two weeks in any given commercially harvested shellfish area, does not completely eliminate biotoxin risk [26].

In comparison to commercial harvests, prevention of self-harvesting illnesses in indigenous communities and for recreational harvesters has been less successful. Misinterpretations of posted shellfish closure signs have been described as playing a role in shellfish toxin related illnesses, arising when signs posted in recreational areas provided seasonal sanitary closure dates, but did not specify biotoxin risks or whether biotoxin tests were conducted in that area. Signage in community and recreational areas that clearly communicate current biotoxin risks in a given area that include clear opening and closing dates and provide sanitary and/or biotoxin warnings would improve public risk communication and awareness in areas where shellfish harvesting is occurring. Currently, shellfish closure data provided by federal authorities are interpreted at maps.bccdc.org/shellfish, but biotoxin data are not provided as they are not publicly available. Better public awareness, education, and access to online resources of information for shellfish closures and self-harvesters are required.

Recent research found biomonitored species (mussels) do not reflect the consumption risk in other shellfish species (scallops) on the east coast of Canada [46]. As was seen in several BC investigations as well, biomonitored species did not accurately reflect risk in other shellfish species. The length of time in which shellfish bind toxins is dependent on the species and toxin source created by phytoplankton species [47]. *Mytilus* spp. (mussels), the most common species for biomonitoring, can lose bound toxins within days and littleneck and manila clams within five weeks, and butter clams can retain toxins for over two years [47]. Despite the important health implications of these large differences in toxin-retention times, self-harvesters in five investigations over the past 10 years were not able to identify the species of clams they ingested even though the areas they harvested from were partially open to some species, such as manila and littleneck clams, but closed to others, such as geoducks and butter clams. Further, biomonitored samples may not detect toxins when present. Early research in BC by Quayle in the 1960s found individual butter clams sampled and tested in close proximity to each other from the same beach ranged from 50 to 1568 µg STX-eq/100 g [9].

Similar to reports from Washington and Alaska [3,5,36], clams of various species were most often associated with PSP in BC, followed by mussels and oysters. Butter clams (*Saxidomus giganteus*) were the most frequently identified (42%) species and are high risk for PSTs because they can take a year or longer to eliminate toxins from tissues after exposure to toxic dinoflagellates [44]. Until recently, there were no reports of PSP linked to crabs in BC. Two cases since 2006 linked to consumption of whole crabs had mild symptoms and recovered. No harvest area was recorded for one case, and in the other, biomonitored mussels from the same harvest area were below regulatory thresholds (44 µg/100 g STX-eq). One case in Alaska in 2010–2011 who consumed crab and its viscera appeared to recover, then died the following day, although this death was attributed to heart disease [34]. Toxins are found and regulated in Dungeness crab fisheries in Washington State [48]. Unlike the US, Canada does not test for, nor does it have regulatory limits established for, phycotoxins (PSTs, domoic acid, dinophysis, or other harmful toxins) in crabs or other invertebrates (e.g., prawns). In the absence of regulatory limits, it is important to communicate the risks involved with cooking and consuming crabs. PSTs and other phycotoxins are heat-stable and potentially become more toxic following cooking [10]. Toxins accumulated in the hepatopancreas of crabs can contaminate cooking water, broth made from cooking water, and the crab flesh itself. Consumers should be educated to not cook crabs whole as a safety precaution; instead, crabs should first be killed, split in half, and have the visceral contents removed before boiling. Understanding marine foods affected by bio-concentrating of PSTs and other phycotoxins of concern is important to food security of communities reliant on self-harvesting and to their cultural connection to their lands.

Limitations of this study included under-reporting of PSP for several reasons. BC’s poison control center was first established in 1975. Twenty-eight (60%) of documented PSP investigations came via poison control notifications, which may explain why fewer cases could be identified in the first 40 years of this review [49]. Requirements for reporting were only introduced in 2001, when PSP was added to BC’s list of reportable diseases. Although the search strategy for PSP case illness was comprehensive, mild cases of PSP that self-resolved, or may have been misdiagnosed as gastrointestinal illnesses, would likely have gone unreported. Indigenous people are often not seen by health care practitioners unless they suffer severe illness, given mistrust of government and racism in health care facilities. It is possible that mild cases of PSP from indigenous peoples are under-represented because PSP symptoms of drowsiness, slurred speech, and staggering gait have in the past been attributed to alcohol intoxication rather than shellfish poisoning, related to persistent degrading racial stereotypes [50]. Reporting methods, clinical case definitions, and health care practitioner awareness of PSP have also changed over time. Symptom descriptors for PSP were not standardized until online reporting forms were introduced that use check-box formats. Improvements to PSP surveillance in recent years include establishing a reporting mechanism for clinically diagnosed cases through Panorama (a national illness reporting tool used by the province) and establishment of real-time monitoring of shellfish illness reports via poison control centers that are investigated by public health authorities [51]. Criteria used to confirm clinical diagnoses of PSP, marine water dinoflagellate monitoring, and urinary or fecal testing of cases are not readily available to public health practitioners in BC [52]. In rare instances, public health has sent clinical samples to the US for diagnostic testing and confirmation. Urinary PST has been used in Alaska to evaluate clinical probable diagnosis of PSP [34]. PST biomonitoring in bivalves suggests rapid rises in toxicity can occur. Monitoring of *Alexandrium* spp. and environmental factors would be useful to understand why rapid increases in PST occur. We were unable to compare PSP illnesses to phytoplankton bloom occurrence as longitudinal records of bloom events are not available for BC shellfish harvest areas. Additionally, there is no national phytoplankton monitoring program in Canada, making it difficult to assess direct influence of climate change on shellfish toxicity events [53]. Temperature and salinity variations can be localized, suggesting local monitoring for harmful algal blooms is necessary to assess bivalve toxicity [54].

## 4. Materials and Methods

### 4.1. Establishing a Historical Record of Paralytic Shellfish Poisonings in BC

A historical record of shellfish illness reports from 1940 to 2020 was based on information from 14 sources, including published articles, national and provincial serial magazines and newsletters, faxes, letters and e-mails, electronic databases (National Notifiable Diseases Online, Public Health Agency of Canada), electronic surveillance systems, BC Drug and Poison Information Centre (poison control call line) shellfish calls since 1975 and surveillance system initiated in 2015, Fishery Notices published online since 1999, Statistics Canada (health and environment) records, newsprint, and Google©. Search terms included PSP, paralytic shellfish poisoning, saxitoxin, BC, British Columbia, neurological symptoms, seafood, illness, harmful algal blooms, and HABs. The search was conducted up to and including December 2020. E-mail traffic was recorded when it was the principal source of information for the investigation. Duplicate reports from multiple information sources were excluded.

### 4.2. Definitions for Paralytic Shellfish Poisoning Cases and Investigations

A probable case of PSP was defined as clinical illness within 12 h of consumption of at-risk shellfish (bivalves or crab) in the absence of other known causes. Clinical PSP illness was defined by neurological symptoms (paresthesia and/or paralysis involving the mouth and extremities) with or without gastrointestinal symptoms [52]. A confirmed case of PSP included the clinical case definition and detection of PSTs in shellfish consumed in edible tissues OR in biomonitored shellfish from the implicated harvest area in excess of 0.8 mg STX-eq/kg (0.8 ppm) OR detection of PSTs in patient urine or feces collected within 24 h of exposure and illness [52].

PSP investigations were sub-divided into single cases (individual exposures) and clusters involving two or more cases. PSP investigation reviews included epidemiological information from cases to describe case illness within the cluster, types of shellfish consumed, where shellfish were self-harvested or purchased, harvest area status, and laboratory testing of leftover or biomonitored shellfish related to the investigation. Seafood species consumed were counted once for each species represented within an investigation, i.e., if case(s) ate more than one seafood species, each seafood species was recorded once per investigation. Case information was collated from all investigations to enumerate frequency, onset and duration of symptoms, and demographic (age, gender) variables. Due to the time span covered and the inconsistent recording of information over the study duration, case and investigation denominators are reported alongside descriptive data. As with cases, investigations were confirmed with supporting STXs-eq values at or exceeding 0.8 mg STX-eq/kg in shellfish, and probable PSP status was assigned when values of <0.8 mg STX-eq/kg were detected in leftover and/or biomonitored shellfish from the implicated harvest area, when no shellfish data were available, or when the harvest area for the implicated shellfish was unknown.

### 4.3. Temperature and Seasonal Trends

Temperature was estimated from six SST buoys located along the west coast of BC. The open data set published by Fisheries and Oceans Canada lists monthly averaged SSTs derived from daily temperature measurements. The buoy locations were chosen based on data availability for the period 1940 to 2020 with greater than 80% of records complete. The average of all monthly SST stations data provided annual averages and 10-year period averages for SSTs [55] that were regressed against the number of investigations per 10-year period. Chi-squared analyses were used to compare the proportion of PSP investigations over 20-year periods, the number of PSP investigations per month and season, and in months spelled with and without an ‘R’, to test the misconception that it is safe to harvest shellfish between September to April and not between the summer months of May to August.

## 5. Conclusions

A method to notify stakeholders when HABs are occurring before regulatory limits are reached is needed to prevent PSP illness. Establishing regulatory thresholds for foods known to be at risk for biotoxins, particularly mollusks but also including crabs and other invertebrates, is urgent. Phycotoxins are found in many species other than bivalve mollusks, including gastropods, echinoderms, and crustaceans [12], and elevated levels of PSTs have been detected in North Atlantic species of limpets, sea snails, starfish, and sea urchins [56]. To determine health risk, an updated survey of biotoxin occurrence in Pacific seafoods is warranted as more marine foods are harvested and as climate impacts on these foods become better known. Developing partnerships within and outside of BC that include citizen science ocean monitoring groups, indigenous monitoring groups along the Pacific coast, and national partnerships invested in protecting shellfish sources and ocean health, are important to understanding HAB development and progression. Introducing PST testing and phytoplankton monitoring in self-harvest areas, especially for communities that rely on shellfish as a primary food resource, such as indigenous communities, is also needed. A combined HAB monitoring effort, including Alaska, Washington State, and BC, would allow for a more complete picture of HAB occurrence along the west coast of North America.

## Figures and Tables

**Figure 1 marinedrugs-19-00568-f001:**
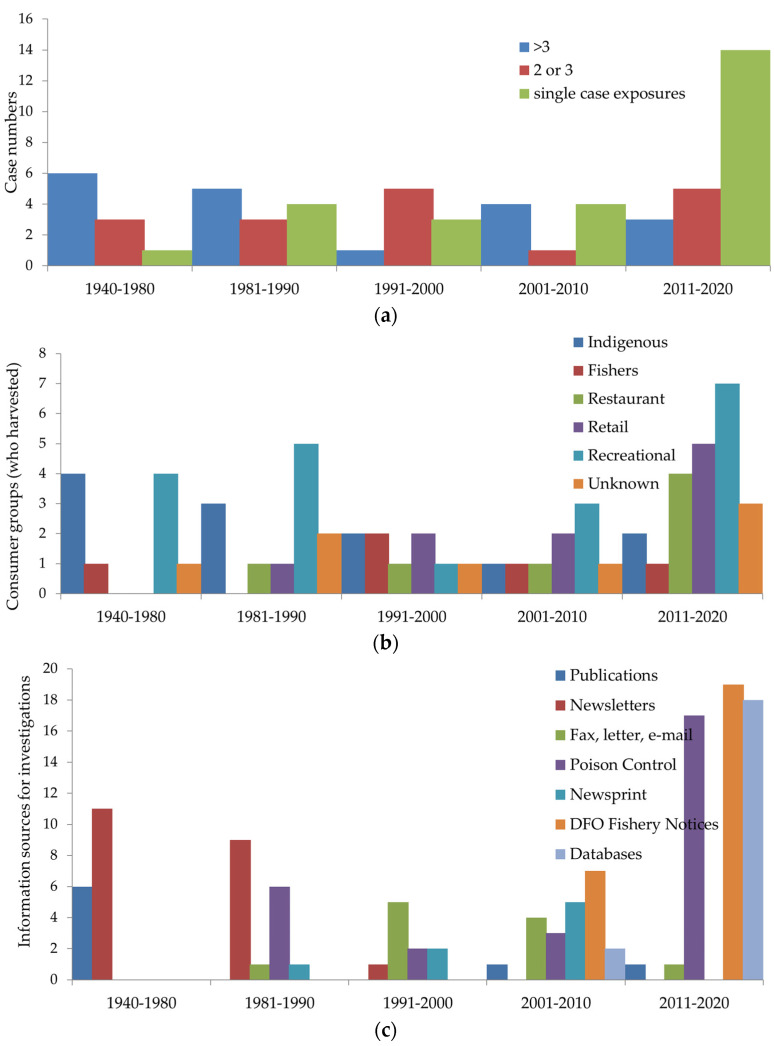
Temporal distributions of investigation evidence by (**a**) number of cases in PSP investigations; (**b**) consumer groups; (**c**) information sources for investigations; (**d**) shellfish species consumed; (**e**) PST amounts detected; and (**f**) quality of shellfish toxin evidence.

**Figure 2 marinedrugs-19-00568-f002:**
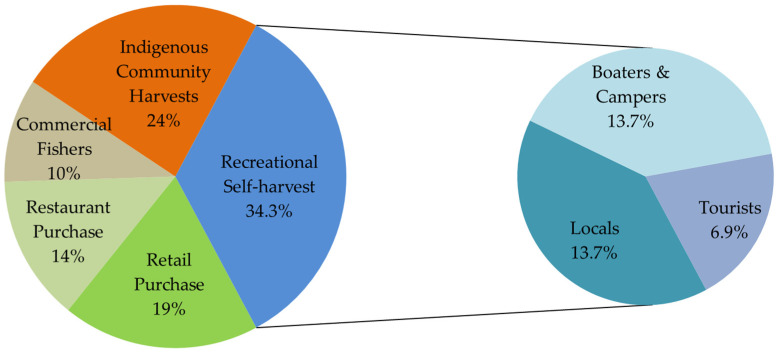
Demographics of paralytic shellfish poisoning investigations in BC, 1940 to 2020.

**Figure 3 marinedrugs-19-00568-f003:**
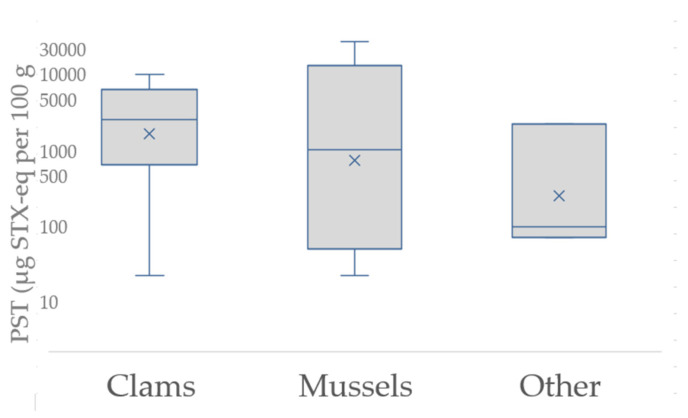
PST in shellfish associated with PSP illness in BC.

**Figure 4 marinedrugs-19-00568-f004:**
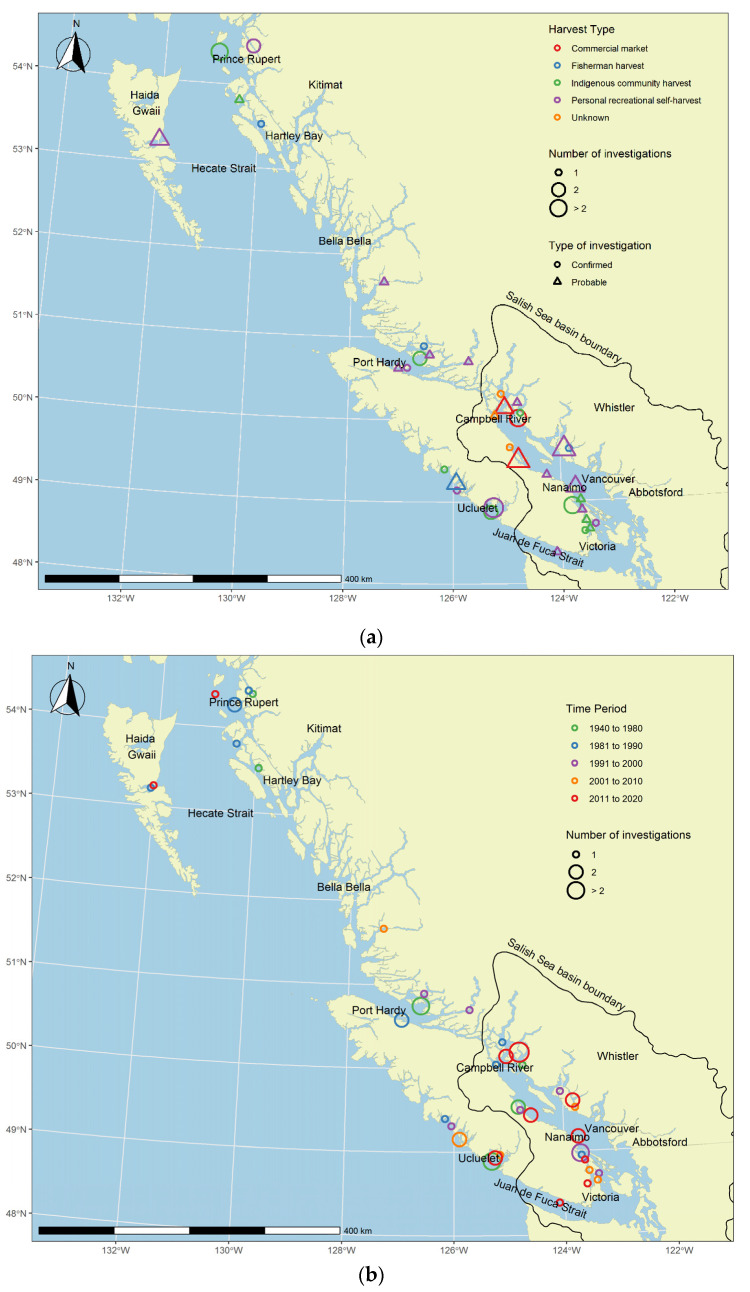
Harvest site locations for shellfish poisoning investigations in BC from 1940 to 2020: (**a**) harvester demographics; (**b**) temporal distribution.

**Figure 5 marinedrugs-19-00568-f005:**
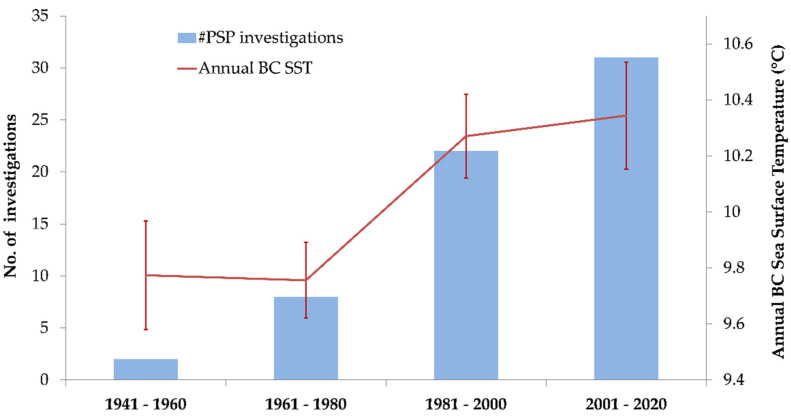
Paralytic shellfish poisoning (PSP) investigations and sea surface temperatures between 1941 and 2020 in BC, Canada.

**Table 1 marinedrugs-19-00568-t001:** Epidemiological summary of case information.

	Sex	Age (yrs)	Onset (hrs)	Duration (hrs)	Symptoms
	Male 56.7%	Average 43.7	Average 3.2	Average 38.3	Perioral tingling 74.1% and perioral numbness 20.6%
	Female 43.2%	Median 46	Median 1.5	Median 24	Numbness 68.3%
		Range 8–91	Range 0.08–18	Range 0.5–252	Extremities hands, finger, legs, and toes tingling 45.5%, and numbness 19.6%
					Ataxia 42.9%
					Paralysis 18.0%
					Abdominal cramping 15.3%
					Floating, dizziness 14.3%
					Nausea 13.8%
					Vomiting 13.8%
					Weakness 9.5%
					Diarrhea 7.4%
					Headache 5.8%
					Difficulty breathing 4.2%
					Swelling in lips, tongue, or face 3.2%
					Dysphagia/dysarthria 2.6%
					Chest pain or rapid pulse 2.6%
					Hot, sweaty, and feverish 2.6%
					Ptosis 2.1%
					Loss of consciousness 1.6%
Summarized from no. of cases	111	61	97	102	189

**Table 2 marinedrugs-19-00568-t002:** Harvest area status of the shellfish sources implicated in confirmed and probable paralytic shellfish poisoning investigations.

	Self-Harvested		Commercial	
Harvest Area Status	Confirmed	Probable	Confirmed	Probable
Open	7	6	0	8
Closed	13	13	2	0
Unknown	2	3	0	7

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
