# Peer review of "Changing Trends in Paralytic Shellfish Poisonings Reflect Increasing Sea Surface Temperatures and Practices of Indigenous and Recreational Harvesters in British Columbia, Canada"

_marinedrugs, 2021, doi:10.3390/md19100568_

Round 1
Reviewer 1 Report
Review of 'Changing trends in paralytic shellfish poisonings reflect both climate change and evolving practices of indigenous and recreational harvesters in British Columbia, Canada' submitted to Marine Drugs.
This study collected reports of SXT-eq poisonings from 1940 onwards, compared occurrence with sea surface temperature, and analysed to gain a better understanding of the socio-economic influences of SXT-eq poisoning events.
This is a well resourced dataset and provides interesting insight into the occurrences of SXT-eq poisoning events in BC.
a few minor comments:
Line 88: the first sentence of the results is a bit confusing, it reads as if the authors collected data themselves from 1940. It would be clearer as: 'We collated reports of 62 PSP investigations between 1940 and 2020.
Fig 1. e. the legend of the y-axis is misleading: 'shellfish samples with STX' would be clearer.
Line 235: I was initially confused with the reference to the 'R' months, but it is explained in the discussion.
Line 246 - 247: the first sentence of the discussion is missing a comma after Canada.
Reviewer 2 Report
This manuscript collected data on PSP toxins concentration in shellfish from British Columbia, Canada, and data from food poisonings to associate their increasing occurrence with climate change (increase of sea surface temperature). The authors also aimed to associate food poisonings with people’s behaviour and activities.
In my opinion important data is here presented, but this manuscript is more a descriptive report than a research article.
The title of manuscript is rather speculative. The relationship shown in the present study associating increasing PSP occurrence with increase of SST is vague. Deeper analyses are needed to effectively reach this observation. Is this SST the average value for each year? What are the variability? Why not plotting each data of PSP against the corresponding SST?
Legend of figure 1 is difficult to read.
Figure 2 is difficult to understand.
I suggest an extensive re-writing, and deeper analyses of these data to reformulate and revise the manuscript.
Reviewer 3 Report
General
This manuscript is a good pioneering study given the still current PSP incidence not only in Canada but in many other countries. Even if the confirmation of PSP intoxication was limited in some cases, the authors did mention the methodology to classify the poisoning as PSP and the potential limitations of their study. I provide some particular comments to improve the study.
Particular
Line 22-24. It could also help to add “and to determine the bloom season or period when PST-producing algae are present at high concentrations”
Lines 35, 49. The term STX-eq is mostly used for quantification and reporting purposes of PST. It is preferred to use the term paralytic shellfish toxins or PST in the entire manuscript instead of STX-eq.
Line 37. Gonyaulax has been reassigned as Alexandrium, so “Gonyaulax spp.” should be removed.
Lines 40-41. The term HABs is more general than the way it is employed in this sentence, and it’s not particular to STX-producing microalgae. These sentences need to be rewritten.
Lines 49-50. Please review the main mechanism of action of PST, focus more on sodium channels.
Lines 68-71. Reference needed for that statement.
Line 78. This is when the usual reporting units should be used: 0.8 mg STX-eq/kg
Figure 1. The size of the figure and graphs are very small, especially the legends. Please increase the font size as well as column sizes.
Lines 120-124. It would be interesting to know whether the earlier reports (pre 1980) were properly diagnosed as PSP. In many cases, toxin-shellfish poisonings are misdiagnosed due to the lack of information or knowledge of these phenomena. Do hospitals and clinics have any access to this information and events nowadays? Given that indigenous communities have had the greatest PSP incidence and shellfish consumption might be part of their traditions, do they have access to this information and education about PSP? Do harvest areas have warning signs during bloom seasons?
Figure 2. Please increase the font size.
Lines 176-178. In order to maintain uniformity, please provide the units as STX-eq/mg (as previously mentioned, and not in log units).
Figure 3. Similar to other figures, they are too small. Please increase the size, including font size.
Lines 230-241. Is there any information about the main phytoplankton species causing the blooms? According to the toxin testing results available and given that clams and mussels have been the main vector for PSP events, do these two species accumulate the toxins at higher levels? It would be important to include the toxin levels found in each shellfish type. I see the table in supplementary information contains some of this information (or at least as confirmed); however, it would be better to present a graph or table with those shellfish species and PST levels found across the study period.
Lines 306-322. Were there any testing using rapid kits? The results can be known within minutes. The importance of introducing routine testing with rapid kits can be discussed here, and in the conclusion.
Lines 329-336. This answers part of my points raised above. How about hospitals/clinics, are they also informed about this?
Line 359-360. Uniformity of reporting units.
Lines 427, 441-442. Corrects units to 0.8 mg STX-eq/kg
Round 2
Reviewer 2 Report
The authors made extensive changes to the manuscript. I can now recommend it for publication in Marine Drugs.